# Ethylene Action Inhibition Improves Adventitious Root Induction in Adult Chestnut Tissues

**DOI:** 10.3390/plants13050738

**Published:** 2024-03-06

**Authors:** Ricardo Castro-Camba, Mariana Neves, Sandra Correia, Jorge Canhoto, Jesús M. Vielba, Conchi Sánchez

**Affiliations:** 1Department of Plant Production, Misión Biológica de Galicia, CSIC, Avda de Vigo s/n, 15705 Santiago de Compostela, Spain; ricardo.castro@mbg.csic.es (R.C.-C.); jmvielba@mbg.csic.es (J.M.V.); 2Centre for Functional Ecology, TERRA Associate Laboratory, Department of Life Sciences, University of Coimbra, Calçada Martim de Freitas, 3000-456 Coimbra, Portugal; mariananevespt@gmail.com (M.N.); jorgecan@uc.pt (J.C.); 3InnovPlantProtect CoLab, Estrada de Gil Vaz, 7350-478 Elvas, Portugal

**Keywords:** auxin, *Castanea sativa*, ethylene, maturation, recalcitrance, root induction

## Abstract

Phase change refers to the process of maturation and transition from the juvenile to the adult stage. In response to this shift, certain species like chestnut lose the ability to form adventitious roots, thereby hindering the successful micropropagation of adult plants. While auxin is the main hormone involved in adventitious root formation, other hormones, such as ethylene, are also thought to play a role in its induction and development. In this study, experiments were carried out to determine the effects of ethylene on the induction and growth of adventitious roots. The analysis was performed in two types of chestnut microshoots derived from the same tree, a juvenile-like line with a high rooting ability derived from basal shoots (P2BS) and a line derived from crown branches (P2CR) with low rooting responses. By means of the application of compounds to modify ethylene content or inhibit its signalling, the potential involvement of this hormone in the induction of adventitious roots was analysed. Our results show that ethylene can modify the rooting competence of mature shoots, while the response in juvenile material was barely affected. To further characterise the molecular reasons underlying this maturation-derived shift in behaviour, specific gene expression analyses were developed. The findings suggest that several mechanisms, including ethylene signalling, auxin transport and epigenetic modifications, relate to the modulation of the rooting ability of mature chestnut microshoots and their recalcitrant behaviour.

## 1. Introduction

The European sweet chestnut (*Castanea sativa*) is a profitable and cost-effective woody species in the Mediterranean basin [1,2], highly relevant for biomass, timber and fruit production [3,4]. Chestnut trees have a great ecological value, as they have been related to an increase in plant species richness, and they are important carbon sinks aiding to mitigate climate change [5,6]. In addition, chestnut trees are involved in the maintenance of traditional landscapes, significantly contributing to the environmental and cultural heritage [7,8]. Nonetheless, their potential applications are hindered by their recalcitrant behaviour, thus driving the use of biotechnological approaches to overcome this limitation. Selected chestnut genotypes have been successfully micropropagated using both juvenile and mature material, although acceptable rooting rates have been attained only in juvenile-like material (reviewed in [9]). However, rooting ability is also greatly influenced by the genotype, thus limiting the pool of available material for the vegetative propagation of this species.

Adventitious rooting (AR) is a complex post-embryogenic process through which roots are formed in tissues not previously determined to form roots. AR is modulated by several internal factors such as mother plant status, the chronological and ontogenetic age, the genetic makeup, and hormonal balance, as well as external conditions like temperature, light and mineral nutrition [10,11,12]. This process is a common occurrence in plant development and can be triggered as a response to different stressors such as flooding, nutrient deficiency, wounding or oxidative stress [13,14]. The formation of AR is a key step in the vegetative propagation of plants. Many species, including several trees, exhibit a recalcitrant behaviour, and their ability to form roots decreases dramatically with age as they go through the phase change from the juvenile to the mature stage, with low responses to rooting stimuli [11,15,16]. The reversion of maturity-related traits, particularly the improvement of AR, represents a challenge for the propagation of species whose desirable traits are seen only in the mature stage. The lack of rooting or the deficiencies in root architecture reduce the survival rate of plants and hamper the mass propagation of selected genotypes, causing heavy economic losses for the related industries.

AR is divided into three sequential stages, named induction, initiation and expression (outgrowth). In chestnut, the most limiting stage is the induction phase, in which specific cells respond to stimuli and initiate a root developmental program in root-prone tissues/genotypes, which takes place in the first 24–48 h after the beginning of the induction treatment [9,17]. Every step in the AR process is subject to a dynamic and specific hormone modulation, although auxin is the key hormone in AR [18]. The decline of the rooting ability after the transition from the juvenile to the mature stage is linked to changes in auxin homeostasis [11,16], modifying the expression of auxin-responsive genes [19,20]. However, other plant hormones are involved in the regulation of AR. Plant growth regulators such as cytokinins [21] or gibberellins [22] seem to inhibit AR, whereas jasmonates [23,24] and abscisic acid [25] are AR stimulators in some species. Nonetheless, the roles played by different hormones seem to have a species-specific component, while at the same time, their function might be dependent on the particular stage of the process.

Ethylene (ET) is a gaseous plant growth regulator involved in different plant developmental processes, including close links to plant aging and phase change [26,27,28]. Moreover, ET has been shown to interact with auxin in primary root development and other rooting processes [29,30]. In chestnut, a transcriptomic analysis revealed that signalling related to different hormones such as abscisic acid or ethylene is upregulated in mature tissues in response to auxin and wounding when compared to juvenile tissues. Among them, ethylene signalling genes were found, which led to the hypothesis that higher concentrations of ET in mature tissues could be one of the causes underlying its recalcitrant behaviour [20]. ET biosynthesis and signalling pathways have been properly characterised in plants, leading to the development of specific strategies that allow for their modulation. For instance, ET biosynthesis might be enhanced or reduced by the application of its precursor 1-aminocyclopropane-1-carboxylate (ACC) or aminoethoxyvinylglycine (AVG), respectively. On the other hand, the use of silver ions allows for the inhibition of ET perception (reviewed in [31]). To test the effects of ET on the root system development of chestnut, ACC, silver nitrate (AgNO_3_) and AVG were applied to the root induction media of juvenile and mature chestnut microshoots. The results obtained prompted us to further characterise the responses in mature chestnut tissues, where specific gene expressions related to ET synthesis (*CsACS1-like* and *CsACO1-like*) and signalling (*CsEIN2*, *CsERF3* and *CsRAP2.12*), auxin responses (*CsPIN1* and *CsIAA29*) and epigenetic processes (*CsHDA14* and *CsJMJ30*) were evaluated via quantitative PCR. The results show a negative effect of ET on the induction of AR in mature tissues and a link to recalcitrance, which might at least partially exert its action through interaction with auxin transport and specific epigenetic processes.

## 2. Results

### 2.1. Rooting Experiments

#### Ethylene Negatively Modulates Adventitious Rooting and Worsens Root System Development in Mature Chestnut

Rooting traits were maintained constant throughout all the experiments in the rooting-competent juvenile line (P2BS, Figure 1 and Figure 2), and the rooting rates were 100% for all treatments. Similarly, the average root number per explant and root length parameters were not significantly affected by the treatments. Root length ranged from 3.3 cm in Indole Butyric Acid (IBA)-treated shoots to 4.5 cm in IBA + AVG treated shoots, with intermediate values for the IBA + AgN0_3_ and IBA + ACC treatments (4.3 and 3.7 cm, respectively). However, ET perception inhibition seemed to have a slight effect on the rooting performances of P2BS shoots, decreasing the root number per shoot (Figure 2c) and increasing the root length (Figure 2b). On the other hand, the AVG treatment promoted lateral root formation, while AgNO_3_ reduced shoot tip necrosis (Figure 1c,d).

In contrast, the rooting performance of P2CR shoots was affected by ET modulators (Figure 1 and Figure 2). Shoots induced to root in root induction medium (RIM) exhibited a 27% rooting rate; the lowest rooting percentage (18%) was obtained in ACC-treated shoots, while the maximum rooting percentage was achieved in AgNO_3-_ and AVG-treated shoots (44% and 40%, respectively; Figure 2a). Regarding the root number per explant, ET attenuation improved this trait, more than doubling the data obtained from the IBA and IBA + ACC treatments (Figure 2c). These data highlight a negative effect of ACC in the number of roots per explant, not only in their ability to root. Moreover, the average root length in IBA + ACC treated shoots was 1.7, while it ranged from 6.6 to 7.9 cm in the rest of the treatments (Figure 2c). Interestingly, P2CR shoots rooted in the presence of silver nitrate or AVG showed a lower amount of callus tissue than those rooted in the other treatments, as well as a marked increase in lateral roots, suggesting a long-lasting effect of ET-inhibition in specific root traits (Figure 1g,h). Altogether these results show that the presence of ACC seems to negatively modulate the induction and development of AR in mature shoots, whereas lowering ET action by AgNO_3_, or AVG has the opposite effect.

### 2.2. Analysis of Gene Expression during AR in Rooting-Recalcitrant Shoots

#### 2.2.1. Ethylene-Related Genes

To confirm the effects of ET modulators on the expression patterns of ET-related genes during the AR process, expression analyses of the *CsACS1-like*, *CsACO1-like*, *CsRAP2.12*, *CsERF3* and *CsEIN2* genes were performed in the basal parts of the P2CR shoots, where the rooting response might take place.

The IBA treatment decreased the expressions of both ET synthesis-related genes (*CsACS1-like*, *CsACO1-like*) at 24 h (Figure 3a,b). However, the expression of *CsACS1-like* of the IBA-treated shoots at 120 h was similar to that in T0 samples, while it remained low for *CsACO1-like*. The AgNO_3_-mediated inhibition of the ET perception of IBA-treated shoots did not affect the expression levels of those genes, suggesting that the treatments (IBA, IBA + AVG and IBA + AgNO_3_) restrain ET synthesis to a similar degree (Figure 3a,b). On the other hand, ACC application upregulated the expression of both genes at 24 h, with a sustained induction only for *CsACS1-like*, while at 120 h, the expression of *CsACO1-like* in ACC-treated shoots was lower than in the T0 samples (Figure 3b). These results suggest a positive feedback of ACC on its own synthesis, which was active for 120 h, as well as the promotion of the conversion of ACC in ET in the short term. Therefore, distinct treatments affected the ET biosynthesis pathway differently, which might be related to the dissimilar phenotypic responses found.

To further characterise ET-related responses, the expressions of genes linked to ET signalling were analysed. *CsEIN2*, a core gene controlling gene expression in response to ET, showed contrasting results in response to the treatments. During the first 120 h, the expression of *CsEIN2* was downregulated by the IBA + AgNO_3_ treatment, thus showing a block in ET signalling. On the other hand, at 24 h, the ACC treatment induced the expression of *CsEIN2* in IBA-treated samples, while at 120 h, it did not affect the transcription of *CsEIN2* since the mRNA levels were similar to those detected in the control samples (Figure 3c). IBA alone slightly reduced the expression of *CsEIN2*. Therefore, the *CsEIN2* expression was responsive to ET content-modifying treatments, particularly at 24 h. Two transcription factors involved in the ET responses were also analysed. At 24 h, the expression of *CsERF3* decreased in response to AgNO_3_ and AVG, while it was scarcely induced by ACC (Figure 3d). In the rest of the samples, no significant shifts were detected in the expressions of the genes. These results suggest that the *CsERF3* expression is determined by ET content in chestnut mature tissues, showing a temporary response that might be related to early events in AR. On the other hand, the expression of *CsRAP2.12*, an ET-responsive transcription factor from group VII, showed no significant changes in response to the different treatments. However, all samples showed a lower level of expression than the T0 one, except for the ACC-treated 24 h samples (Figure 3e). These results suggest that *CsRAP2.12* is modulated by ET at 24 h; however, in the rest of the samples and time points subjected to analysis, the transcription of the gene was not affected by the IBA and ET-related treatments.

#### 2.2.2. Auxin-Related Genes

Due to the role of auxin in the AR process, an expression analysis of specific genes related to this hormone was performed. Regarding *CsPIN1*, a membrane transporter involved in auxin polar movement, the IBA treatment increased the expression of the gene at 24 h and 120 h, which was expected by the increase in the auxin concentration in the tissues. However, no changes in the *CsPIN1* expression were detected in ACC-treated samples compared to the control (Figure 4b). Strikingly, the expression level of the gene was dramatically induced by the AgNO_3_ and AVG treatments at 24 h, and were maintained upregulated at 120 h (Figure 4a). Therefore, the induction of *CsPIN1*, an auxin transporter gene involved in the generation of hormone gradients, through ET perception inhibition or ET synthesis blocking correlates with the improvement of AR. In the case of *CsIAA29*, no significant changes in gene expression were detected (Figure 4b). Therefore, the improvement of AR in response to ET perception or synthesis inhibition shows close links with auxin-related processes, specifically that of transport.

#### 2.2.3. Epigenetic-Related Genes

Two genes linked to epigenetic responses were analysed to infer possible mechanisms of AR induction in mature tissues and their relation to hormone signalling. *CsHDA14*, a histone deacetylase, showed no significant changes in response to the different treatments (Figure 5b).

*CsJMJ30*, a histone lysine-specific demethylase gene, showed to be responsive, specifically at 24 h after the beginning of the treatment. At 24 h, the expression of the gene was upregulated by the ACC treatment, while at 120 h, the *CsJMJ30* expression levels dropped back to T0 levels. When only IBA was applied to the shoots, no significant changes in the *CsJMJ30* expression were detected (Figure 5a). ET perception and synthesis inhibition significantly repressed the expression of the gene at both time points, suggesting that AR improvement derived from ET blocking might at least partially exert its effect through the inhibition of this gene.

Overall, the results obtained suggest that ET modifying treatments alter the expression of genes related to ET biosynthesis and signalling, with close links to auxin transport and specific epigenetic mechanisms.

## 3. Discussion

The successful development of AR requires a change in the fate of specific cells not previously determined to form a meristem that will eventually drive the generation of the new organ [32]. In the case of rooting-competent chestnut microshoots, cells close to the vascular bundles in the stem are able to respond to the rooting stimuli, auxin and wounding, and then reprogram their ongoing genetic pattern and enter a root developmental pathway [9]. Therefore, the plasticity of those rooting-competent cells allows them to modify their gene expression patterns in response to wound stress and external auxin supply and switch their fate to AR founder cells. However, this ability is drastically reduced during maturation, with adult microshoots showing a recalcitrant behaviour that severely hampers their ability to form roots. Despite recent advances, the molecular basis of this recalcitrant behaviour is still poorly understood. Several players are believed to take part in this connection between AR and recalcitrance, particularly hormones, epigenetic mechanisms and their crosstalk, which may underlie developmental plasticity in plants [33]. A recent analysis showed that transcriptomic responses to auxin and wounding vary greatly between the two types of microshoots used in the present study [20]. Among the differences found, mature shoots exhibit an increased response related to ET, including biosynthesis-related genes and ET-responsive transcription factors. To gain deeper insight into the putative role of ET in the modulation of AR in chestnut and its relation to recalcitrance, treatments were designed where the ET content was either increased or decreased, or its perception was blocked. In juvenile shoots, no significant phenotypical changes were detected, suggesting that ET does not influence AR in these shoots. However, wounding is a necessary step for AR induction in cuttings and microshoots [9,10], and injury-related stress induces a temporary increase in ET and jasmonic acid (JA). These two hormones develop an antagonistic relation in which JA stimulates the expression of wound-responsive genes, while ET blocks that expression in order to locally and temporarily restrict the repairing response [34]. However, the lack of an ET effect in P2BS shoots suggests that other mechanisms might be active in these tissues to drive tissue repair. On the other hand, ET modulation severely impacted the rooting response of mature shoots, including rooting rates, root number and root length. Therefore, our results suggest that repairing mechanisms and root induction processes might be different according to the ontogenetic state of the tissues.

ET has been shown to present contrasting effects in the formation of AR in different species. It was described as an AR inhibitor in peach [35], Eucalyptus [36] and *Malus x domestica* [37]. On the other hand, there are species such as petunia [38], cucumber [39], marigold [40] and woody plants such as *Pinus thunbergii* [41] and *Citrus sinensis* [42] in which ET stimulates AR formation. However, the effect of this hormone in AR has not been analysed regarding the ontogenetic state of the tissues in woody species. In Arabidopsis, ET-related signalling was shown to be more active in old leaves, which was linked to a low ability for de novo root regeneration [43]. Therefore, ET’s influence on regeneration processes and particularly on AR might be directly connected to the age and development of the plants.

A remarkable effect of ET inhibition in P2BS and P2CR shoots was the ability to induce lateral roots (Figure 1), a trait that improves the root system performance. The formation of these lateral roots took place long after the AgNO_3_ and AVG treatments ended, thus suggesting a lasting effect on the microshoots’ performances. The inhibition of lateral root formation by ET has also been described in Arabidopsis [44,45] and tomato [46]. The possible mechanism of this effect in our system is unknown, although a role for auxin transport modulation might be a plausible explanation (see below).

According to the results of the present study, ET signalling is involved in the induction of AR, at least in mature shoots. ET content and signalling modulation were shown to directly influence genes involved in ET synthesis. Particularly, ACC-treated P2CR shoots showed significant increases in the expressions of *CsACS1-like* and *CsACO1-like*, and these increases were related to a lower rooting response. Surprisingly, ACC triggered a positive feedback loop in the expression of *CsACS1-like* that lasted up to five days after the beginning of the treatment. This gene codes for an enzyme that catalyses the conversion of S-adenosyl-l-methionine (SAM) into ACC, which is later transformed into ET through the activity of ACO enzymes. Thus, in mature shoots, ACC induced an increase in its own synthesis, whose presence seems to prevent specific developmental processes, at least in this system. In recent years, several reports have suggested that ACC might exert signalling effects on its own, not only due to its role as an ET precursor. Those activities have been related to developmental processes and stress responses, and putative transporters for ACC have been identified [47,48]. However, the ACC effect on the expressions of ET-related genes suggests that this might be its main route of action in the system used here.

On the other hand, the IBA, IBA + AVG and IBA + AgNO_3_ treatments reduced the expressions of ET synthesis genes, particularly at 24 h, thus indicating that lowering ET synthesis is necessary to induce rooting responses in mature shoots. The expression of *CsEIN2*, a core component of ET signalling, was modified according to the treatments, with greater levels of expression in samples treated with the precursor of ethylene biosynthesis, while lower levels were detected when ET perception was inhibited. EIN2 acts downstream of ET receptors and modulates the activity of ET-responsive transcription factors (ERFs), also integrating inputs from other pathways in its expression. Here, *CsEIN2* mRNA levels are clearly related to the rooting behaviour of mature shoots, with improved rooting responses by blocking the ET signalling, and thus decreased the *CsEIN2* expression. In Arabidopsis, an ACC + IBA treatment reduced the number of ARs, with the antagonistic relation between *AtEIN2* and the JA signalling gene *AtCOI1* showing to be particularly relevant in response to the IBA induction [49]. Moreover, EIN2 also seems to control ET-related gene expression by inducing histone acetylation, in what is suggested to be a rapid transcriptional regulation process [50]. Our data support this idea of a fast and dynamic control because *CsERF3* showed a parallel expression pattern when compared to *CsEIN2*. However, previous reports suggested a positive role of *ERF3* in the formation of AR in Populus under normal and low-phosphorus conditions, which seems not to match the results in our system, and a link to auxin signalling [51,52]. Probably, ET-related variations in the control of gene expression between both species underlie the differences found.

On the other hand, the expression of the ET-responsive *CsRAP2.12* transcription factor also slightly resembled the *CsEIN2* expression pattern at 24 h. Previously, Valladares et al. [53] analysed the expression of this gene in chestnut and oak tissues during AR induction. The authors suggested that it might be implicated in the establishment of new developmental programs in an ontogenetic-related mode. This gene belongs to group VII of ERFs, of which their activity has been linked to specific responses like low-oxygen conditions or oxidative stress [54]. Here, the levels of expression were lower than those in T0 samples except for those of the ACC + IBA treatment at 24 h; therefore, its expression does not seem to relate to improved rooting responses.

ET and the key AR inducer, auxin, are known to interact in many processes and at different levels. Particularly, ET has been suggested to influence auxin movement through the modulation of the expression of auxin transporters [55]. The activity of these transporters is essential for the generation of hormone gradients in the tissues, which eventually trigger the process of AR [56]. ET perception and synthesis inhibition dramatically induced the expression of *CsPIN1* at 24 h, while its induction was much more modest in IBA-treated shoots. Thus, ET seems to block auxin transport in mature chestnut tissues, preventing the establishment of hormone gradients needed for the induction of specific developmental responses. The activity of PIN transporters seems crucial in the early steps of AR for the successful outcome of the process, as seen for example in apple and tea nodal cuttings [57,58]. In other experimental systems ET also seems to influence regeneration processes by influencing the auxin distribution, as seen in de novo shoot organogenesis in tamarillo [59].

In response to the establishment of auxin gradients, a specific related gene expression is triggered. The expression pattern *CsIAA29* resembled that of *CsPIN1* at 24 h, when AR induction is taking place. Aux/IAA proteins are auxin responsive and work as repressors of Auxin Responsive Factors, modulating their activity [60]. Therefore, *CsIAA29* might be involved in AR induction, playing a role in the fine-tuning of the auxin signalling events, at least in chestnut. Arabidopsis *AtIAA29* has been suggested to be involved in the modulation of root system development [61], and ET was shown to target this gene through the EIN3 proteins, which work downstream of EIN2 [62]. Therefore, ET negatively influences AR induction in mature chestnut shoots by putatively preventing the establishment of auxin gradients and impeding further related signalling.

As previously mentioned, EIN2 influences histone acetylation. P2CR rooting might also be blocked through epigenetic mechanisms governed by ET, and the present results support this hypothesis. However, *CsHDA14*, a histone deacetylase that reduces the accessibility of the transcriptional machinery to DNA, showed no significant differences among treatments despite a higher level of expression in IBA + AgNO_3_ treated samples. On the other hand, the expression of *CsJMJ30* clearly resembled the ET signalling status of the tissues, with expression being upregulated in ACC-treated shoots and downregulated by the inhibition of ET perception or synthesis. JMJ30 is a histone demethylase that generally acts in conjunction with its paralog JMJ32. In Arabidopsis, *AtJMJ30* has been shown to lead to callus formation by inducing the genes *AtLBD16* and *AtLBD29* [63]. Moreover, it seems to control root elongation in response to abscisic acid [64]. In a previous report, abscisic acid-related gene expression was found to be more active in mature shoots than in juvenile shoots of chestnut [20], thus suggesting that *CsJMJ30* might be integrating different cues into its expression that eventually relate to the recalcitrant behaviour of P2CR shoots, as low levels of activity of this demethylase might be increasing the accessibility of the transcriptional machinery to genes, of which their expressions are needed for the development of AR.

## 4. Materials and Methods

### 4.1. Plant Material and Culture Conditions

Microshoots of P2BS and P2CR lines established *in vitro* from the basal shoots and crown branches, respectively, of an 80-year-old *C. sativa* Mill. tree [65] were used in this study. Shoot cultures were grown in GD [66] culture medium supplemented with 0.1 mg L^−1^ benzyladenine, 30 g L^−1^ of sucrose and 7 g L^−1^ of Bacto Agar as the gelling agent. The culture medium pH was adjusted to 5.6–5.7 and autoclaved at 121 °C for 20 min. Every 4 weeks, the basal callus from well-developed shoots was removed, and the shoots were used for a new multiplication cycle by sub-culturing them on fresh culture medium, or for rooting experiments.

### 4.2. Rooting Experiments

At the end of the proliferation cycle, microshoots devoid of the callus and basal leaves were used in rooting experiments. The root induction medium consisted of GD medium with a 1/3 of macronutrients concentration, 30 g L^−1^ of sucrose, 7 g L^−1^ of Bacto Agar and 25 µM Indole Butyric Acid. For the rooting experiments, microshoots that were incubated in RIM for 5 days under dark conditions were transferred to an IBA-free RIM medium (REM, root expression medium) for 25 days and cultured under a 16 h light/8 h dark photoperiod with a light intensity of 40 µmol m^−2^ s^−1^ provided by cool-white fluorescent lamps.

To test the effect of ET on adventitious rooting, the induction of roots was carried out in (i) RIM, (ii) RIM supplemented with 30 µM 1-aminocyclopropane-1-carboxylic acid (ACC), (iii) RIM supplemented with 30 µM silver nitrate (AgNO_3_) and (iv) RIM supplemented with 30 µM aminoethoxyvinylglycine (AVG). The rooting experiments were carried out with 6 explants per oval Microbox container (OV80 + OVD80 with white filter, Microbox, Deinze, Belgium), three containers per replicate and three repeats per treatment (6 explants per replication × 3 replications × 3 repeats = 54 explants per treatment). At the end of the rooting cycle, the percentage of rooted shoots, root number per explant and root length were recorded. The normality of the data was tested using the Shapiro–Wilk test of normality, and the homogeneity of the variance was analysed using Levene’s test. Then, the data were analysed using ANOVA or a Kruskal–Wallis test with a post hoc comparison, HSD Tukey’s or Dunn’s test, respectively. These analyses were performed in R software, version 4.2.2 [67].

### 4.3. RNA Extraction and qPCR Analysis

Plant material from P2CR shoots was collected at the end of the multiplication cycle (T0), and 24 h and 120 h after the initiation of the rooting experiments (IBA, IBA + ACC, IBA + AgNO_3_, IBA + AVG). The basal parts (1 cm) of the microshoots were cut off, immediately frozen in liquid nitrogen, and stored at −80 °C until use. Samples were homogenised with liquid N_2_, and the total RNA was extracted using a Quick-RNA^TM^ Miniprep Kit from ©Zymo Research according to the manufacturer’s instructions. The concentration and quality of the total RNA were assessed using a ND-1000 Spectrophotometer (NanoDrop Technologies, Wilmington, NC, USA).

The synthesis of the cDNA was performed from 1 µg of total RNA, using the NZY First-Strand cDNA Synthesis Flexible Pack kit following the manufacturer’s instructions. qPCR primers were designed with the Primer Designing Tool Primer-BLAST based on the sequences of *Quercus suber* and *Castanea mollissima*. Genes that had their expressions analysed via qPCR include *Castanea sativa ACC Synthase 1-like* (*CsACS1-like*), *ACC oxidase 1-like* (*CsACO1-like*), *Ethylene Insensitive 2* (*CsEIN2*), *Ethylene Responsive Factor 3* (*CsERF3*), auxin transporter *CsPIN1*, *IAA29* (*CsIAA29*), *Histone deacetylase 14* (*CsHDA14*) and *Jumonji 30* (*CsJMJ30*). For the analysis of the Ethylene Responsive Factor *CsRAP2.12* and the reference genes for qPCR validation *Actin-2* (*CsACT2*) and *Elongation factor 1* (*CsELF-1*), sequences from a previous analysis were used [53]. The primer sequences are shown in Appendix A. Three biological replicates were included for each qPCR analysis, using NZYSpeedy qPCR Green Master Mix (2×), following the provided instructions and with the samples diluted 10 times. The relative gene expression value was calculated according to the 2^−∆∆CT^ method [68]. The results were expressed as relative values, using as reference the expressions of the genes in the control samples (T0), harvested at the end of the multiplication cycle and prior to any treatment. Data normality and homoscedasticity were tested using Shapiro–Wilk and by Levene’s tests, respectively. Then, ANOVA or Kruskal–Wallis tests were performed, and HSD Tukey’s or Dunn’s tests were used as post hoc comparison tests. These analyses were performed in R software [67].

## 5. Conclusions

The rooting experiments carried out in this study revealed the effect of ET on AR in chestnut before and after phase change. The results show a negative role of ET in the induction of AR in mature chestnut-derived tissues but not in juvenile-like tissues, revealing that ET activity depends on the ontogenetic state of plant material. A molecular analysis revealed the existence of feedback-regulatory mechanisms of ACC on its own synthesis. ET perception inhibition through AVG and AgNO_3_ treatments seems to influence root development in three related ways: direct ET response inhibition through *CsERF3* and *CsEIN2* repression, auxin transport modification through *CsPIN1* enhancement and probable epigenetic changes driven by *CsJMJ30* (Figure 6). More research is needed to elucidate the specific modes of action of AVG, AgNO_3_ and ACC on the activity of *CsJMJ30* and *CsPIN1* to regulate root development and their implication in the aging processes of chestnut. However, the results provided in this study open new potential approaches toward understanding recalcitrance in chestnut and improving protocols for its vegetative propagation. In addition, they allow for a better comprehension of the acquisition of rooting competence and its relationship with maturation.

## Figures and Tables

**Figure 1 plants-13-00738-f001:**
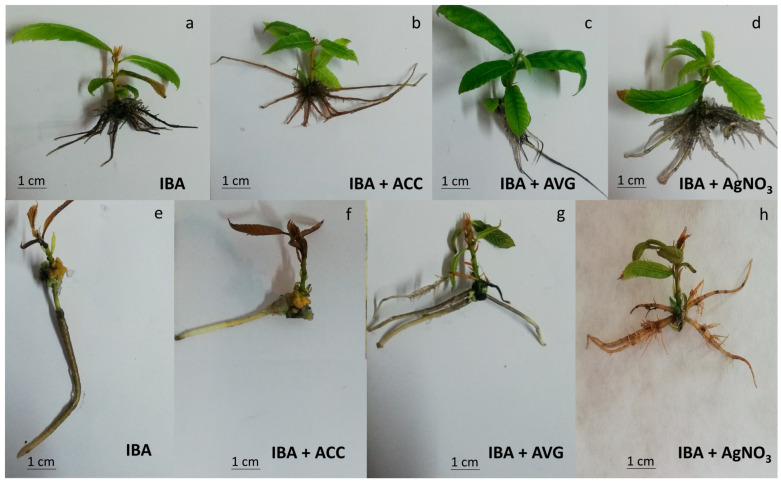
Phenotypes of rooted chestnut plantlets. P2BS shoots are shown in the upper panel after treatment with (**a**) IBA, (**b**) IBA + ACC, (**c**) IBA + AVG or (**d**) IBA + AgNO_3_. P2CR shoots are shown in the lower panel after treatment with (**e**) IBA, (**f**) IBA + ACC, (**g**) IBA + AVG or (**h**) IBA + AgNO_3_. For each experiment 54 microshoots were used (N = 54).

**Figure 2 plants-13-00738-f002:**
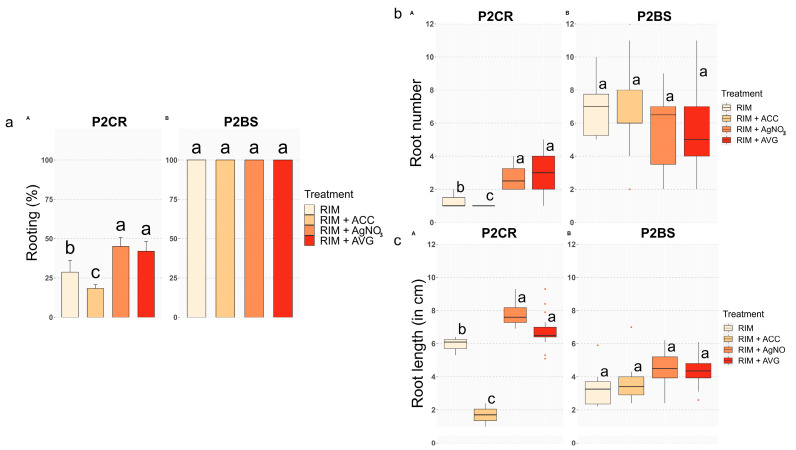
Rooting rates of microshoots from lines P2CR and P2BS. Shoots were treated with IBA 25 µM (RIM), supplemented with ACC 30 µM (RIM + ACC), with AVG 30 µM (RIM + AVG) or with AgN0_3_ 30 µM (RIM + AgNO_3_). Different letters indicate significant differences in each trait (*p* ≤ 0.05). (**a**) Rooting rate; (**b**) Root number; (**c**) Root length.

**Figure 3 plants-13-00738-f003:**
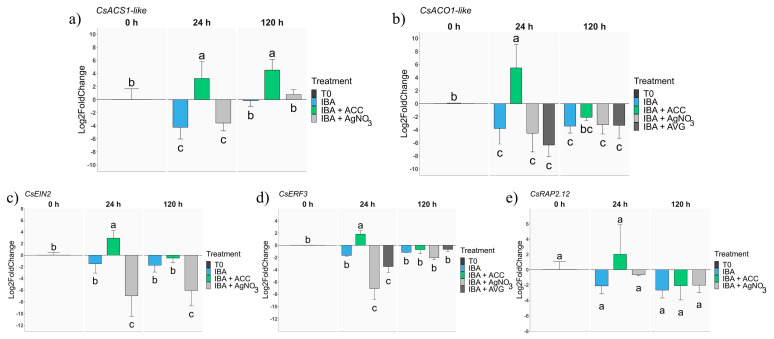
qPCR expression analysis of genes related to ethylene synthesis and signalling. Samples from P2CR shoots were subjected to different treatments: IBA 25 µM (IBA); IBA 25 µM supplemented with ACC 30 µM (IBA + ACC), with AVG 30 µM (IBA + AVG) or with AgNO_3_ 30 µM (IBA + AgNO_3_) and collected 24 h and 120 h after the beginning of the treatments. (**a**) *CsACS1-like*; (**b**) *CsACO-like*; (**c**) *CsEIN2*; (**d**) *CsERF3*; (**e**) *CsRAP2.12*. All data were normalised to the expressions of the genes in the T0 samples. For each gene, different letters indicate statistical differences (*p* ≤ 0.05).

**Figure 4 plants-13-00738-f004:**
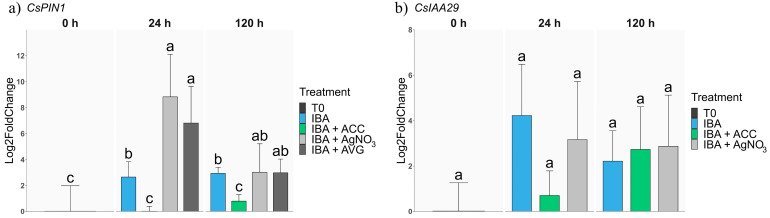
qPCR expression analysis of genes related to auxin transport and signalling. Samples from P2CR shoots were subjected to different treatments: IBA 25 µM (IBA); IBA 25 µM supplemented with ACC 30 µM (IBA + ACC), with AVG 30 µM (IBA + AVG) or with AgNO_3_ 30 µM (IBA + AgNO_3_) and collected 24 h and 120 h after the beginning of the treatments. (**a**) *CsPIN1*; (**b**) *CsIAA29*. All data were normalised to the expression of the T0 sample. For each gene, different letters indicate statistical differences (*p* ≤ 0.05).

**Figure 5 plants-13-00738-f005:**
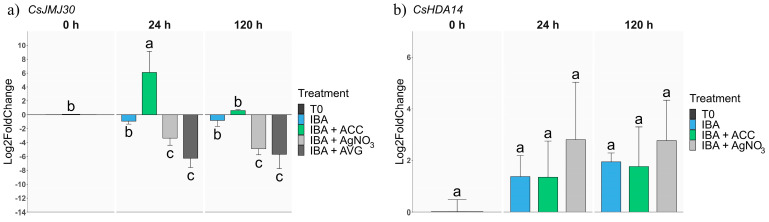
qPCR expression analysis of genes related to epigenetics processes. Samples from P2CR shoots were subjected to different treatments: IBA 25 µM (IBA); IBA 25 µM supplemented with ACC 30 µM (IBA + ACC), with AVG 30 µM (IBA + AVG) or with AgNO_3_ 30 µM (IBA + AgNO_3_) and collected 24 h and 120 h after the beginning of the treatments. (**a**) *CsJMJ30*; (**b**) *CsHDA14*. All data were normalised to the expression of the T0 sample. For each gene, different letters indicate statistical differences (*p* ≤ 0.05).

**Figure 6 plants-13-00738-f006:**
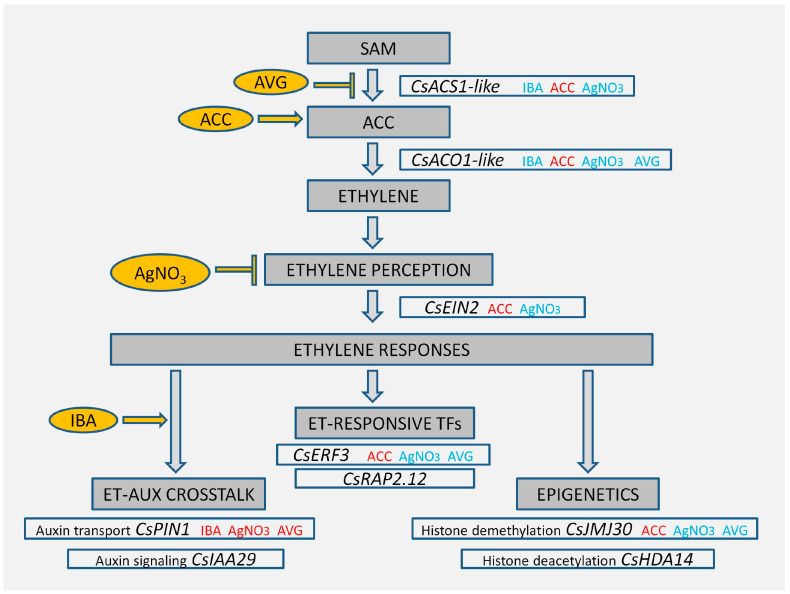
Schematic of the functions of the genes analysed in this study, the treatments applied and the main results from the gene expression analyses. In red and blue, treatments that increased or decreased the expressions of the genes, respectively.

## Data Availability

Data are contained within the article and Appendix A.

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
