# Peer review of "Ethylene Action Inhibition Improves Adventitious Root Induction in Adult Chestnut Tissues"

_plants, 2024, doi:10.3390/plants13050738_

Round 1
Reviewer 1 Report
Comments and Suggestions for Authors
The document refers to the molecular analysis of the effect of ethylene metabolism on root development in chesnuts. The document is of interest, however it requires some modifications to improve.
1. Adjust the photographs of the Figures to the same scale and annotate the scale line appropriately,
2. Improve the gene expression analysis figures,
3. Include a possible scheme of the function of the genes analyzed in the process of induction of roots to favor the reader in the discussion of the results.
4. What about the behavior of the members of the gene families analyzed?
Author Response
Answers to Reviewer 1
Reviewer #1: The document refers to the molecular analysis of the effect of ethylene metabolism on root development in chesnuts. The document is of interest, however it requires some modifications to improve.
- Adjust the photographs of the Figures to the same scale and annotate the scale line appropriately,
Answer: Thank you for the suggestion. We added the suggested changes in order to improve Figure readability.
- Improve the gene expression analysis figures,
Answer: Thank you for your suggestion. Some modifications have been made to the gene expression Figures to improve them. Regarding quality of the images, in the reviewed manuscript we have sent 768 dpi files to ensure high resolution.
- Include a possible scheme of the function of the genes analyzed in the process of induction of roots to favor the reader in the discussion of the results.
Answer: Thank you for your comment. A scheme of genes interaction with plant hormones and their effects on adventitious root inductions was added to the graphical abstract.
- What about the behavior of the members of the gene families analyzed?
Answer: Thank you for your comment. In this study we selected these specific genes based on previous results obtained in a transcriptomic study in our laboratory (Vielba et al., 2022 doi: 10.3390/plants11243486), and were not intended to characterized whole family of genes. In this work we have validated the role of some genes in the induction of ARs in mature shoots.
We would like to thank Reviewer 1 for her/his positive feedback.

Reviewer 2 Report
Comments and Suggestions for Authors
This manuscript compares the roots of juvenile and adult chestnut buds, while treating them with ethylene-related inhibitors or promoters. The authors observed the occurrence of adventitious roots, and molecular differences between the different types of material, which may stimulate further studies on related research topics in plants. However, there are some jobs that need to be added.
1. The authors should add sections to observe and compare the occurrence of root primordia under different treatments to determine the effect of ethylene on roots.
2. The authors should add the changes of endogenous ethylene and auxin in the analysis of ethylene related and auxin related gene changes, which better explain the molecular differences of related ethylene and auxin.
3. The authors analyzed the gene expression patterns associated with epigenetics. However, epigenetics includes DNA methylation, small RNA and chromatin, and the expression of only two genes may not be enough.
4. The authors should introduce the relevant genes in introduction section.
Author Response
Reviewer #2: This manuscript compares the roots of juvenile and adult chestnut buds, while treating them with ethylene-related inhibitors or promoters. The authors observed the occurrence of adventitious roots, and molecular differences between the different types of material, which may stimulate further studies on related research topics in plants. However, there are some jobs that need to be added.
- The authors should add sections to observe and compare the occurrence of root primordia under different treatments to determine the effect of ethylene on roots.
- The authors should add the changes of endogenous ethylene and auxin in the analysis of ethylene related and auxin related gene changes, which better explain the molecular differences of related ethylene and auxin.
Answer: Thank you very much for your comment. The suggestion is interesting, however our results show that ethylene has a relevant although limited influence in the acquisition of rooting competence, as the rooting rate varies from 18% to 40% when ethylene content is modified. Therefore, histological analyses or quantification of endogenous hormones might not provide conclusive results. At the moment, we are testing other hormone modifying compounds with more sound results. When these treatments are optimized, we will carry out the suggested analyses. Therefore, we are deeply grateful for your suggestion and we will take it into account in the line of work opened after these results.
- The authors analyzed the gene expression patterns associated with epigenetics. However, epigenetics includes DNA methylation, small RNA and chromatin, and the expression of only two genes may not be enough.
Answer: Thank you for your comment. It is true that epigenetic changes include other factors associated with chromatin remodeling or methylation. However, the differential expression of a gene encoding a demethylase, as in the case of JMJ30, is an indication of the existence of the action of these epigenetic factors and opens the way to a deeper and more specific work on this type of regulation.
- The authors should introduce the relevant genes in introduction section.
Answer: Thank you very much. Mention has been made of the genes analyzed at the end of the Introduction section (lines 92-96) for consistency throughout the text

Reviewer 3 Report
Comments and Suggestions for Authors
In general terms, the research is well designed and the methodology adequately described. However, the presentation of the results must be substantially improved, particularly, it seems important to me that all treatments are included in the Figure panels. It is especially notable that this is not the case in Figures 3 (panels a, c and e), 4 and 5. Furthermore, it is very important that the description of the results, and consequently the conclusions, are limited to the treatments that show statistical significance. , otherwise it falls into the realm of speculation.

For me, considering that English is not my native language, the manuscript is understandable and well written. It is worth considering some small suggestions, indicated in the attached manuscript.
Author Response
Reviewer #3:
I think ethylene should be included as a keyword
Answer: Thanks for the suggestion. Agree. We included ethylene as a keyword.
These antecedents "predict" the results found (also highlighted), so the novelty of this research is limited (lines 76-79).
Answer: Thanks for your comment. In our previous analysis (Vielba et al., 2022 doi: 10.3390/plants11243486), we compared the gene expression analysis between juvenile-like and mature shoots and the results suggested a role for ethylene in mature shoots rooting recalcitrance. These findings had to be checked through physiological experiments and gene expression, which is what we have done here. Particularly, the influence of ethylene on auxin transport and specific epigenetic processes was not foreseen in the previous work. We believe that this is a major finding. However, other factors were found in the previous work that might be influencing the rooting process, and we are currently analyzing them. However, we reworded the sentence to avoid possible confusion.
IBA is an acronym for?
Answer: Added the complete name.
RIM is an acronym for?
Answer: Added the complete name.
The way the data is presented is very confusing, in all panels the gene expression levels for all treatments should be shown. For example, in panels a, c and e, is the gene analyzed not expressed in the treatment IBA +AVG?, or was not its expression evaluated?
Answer: We agree that the presentation of the figures with a common legend can be misleading. We now present the legends individually since the expression of all genes for AVG treatment was not measured.
p≤
Answer: Change done
The explanation is erroneous since it must be based on statistical significance deferences, according to which two treatments marked with the same letter are equal and should be considered as such. For example, indeed the expression of CsACO1-ike decreases to 24, but that of AcACS1-like does not (which statistically speaking is equal to time 0).
Answer: Thank you for your comment. We changed the letters of the figure in order to be statistically accurate.
These names must be changed in order so that they are according to how they appear in the figure (a for AcACS1-like and be for AcACO1-like.
Answer: Change done
Statistically speaking they are the same.
Answer: Right, we eliminated the sentence as it does not contribute anything to the explanation of the results.
It seems to me that the confusion comes from thinking that if only one letter is shared there are statistical differences, which is not true.
Answer: Thank you for your comment. It is true that the previous notation was erroneous and led to misunderstandings that obscured the results, so the figure has been changed to be more rigorous with the statistical notation.
This statement is not statistically supported.
Answer: Right, we eliminated the sentence as it does not contribute to the explanation of the results.
p≤
Answer: Change done
Again, it would be better if all treatments were shown in both panels.
Answer: We present the legends individually since the expression of all genes for AVG treatment is not measured.
p≤
Answer: Change done
This acronym must be previously described.
Answer: Done.
We would like to thank reviewer 3 for her/his comments that helped increase the quality of the manuscript.

Round 2
Reviewer 1 Report
Comments and Suggestions for Authors
Deficiencies continue to be observed in Figure 1, they must be adjusted to the same scale, including the size of photos..There is no diagram found in the article or in the supplementary material that can indicate the functions of the analyzed genes.
Author Response
Thank you for the suggestions. The Figure 1 has been modified and the scale bar line has been included. A new Figure (Figure 6) has been included in the manuscript with the function of genes
Reviewer 2 Report
Comments and Suggestions for Authors
I hope the author can supplement and improve the content I previously proposed in future research.
Author Response
Dear Reviewer,
Thank you for your suggestions. Future research will address the raised questions